# Quantitative Pose-Based Analysis of Movement Disorders in Paediatric NGLY1 and SLC13A5 Patients

**Chengliang Dai**[1,2]                                                    CHENGLIANG.DAI@UCB.COM
[1] *UCB Pharma Limited, Slough, United Kingdom*
[2] *Imperial College London, London, United Kingdom*

**Phil Scordis**[1]
**Prathyusha Teeyagura**[3]
**Rayann M. Solidum**[3]
[3] *Stanford University, Stanford, CA, United States*

**Jeff Broderick**[4]
**Julia Broderick**[4]
**Jane Broderick**[4]
[4] *Beneufit, Inc., Kentfield, CA, United States*

**Brenda E. Porter**[3]

**Editors:** Under Review for MIDL 2026

## Abstract

Movement disorders have long relied on subjective clinical observation for diagnosis and monitoring. By contrast, modern computer vision tools such as OpenPose can turn video recordings into precise, time-resolved measurements of a patient's posture and movement. In this work, we apply a fully markerless, pose-based pipeline to classify abnormal movements in children with NGLY1 or SLC13A5 mutations. Our primary focus is on simple, physician-informed pose features that can be interpreted in clinical terms and used with conventional classifiers (Random Forest, SVM, etc.) on a very small dataset. We show that these handcrafted features capture clinically meaningful differences between movement-disorder phenotypes and can already achieve useful classification performance. In addition, we include an exploratory comparison with a transformer model that is pre-trained on large-scale action-recognition data and then fine-tuned on our pose data. This experiment illustrates the potential performance ceiling of deep learning with extensive pretraining, but we emphasise that such models are less transparent and more data-hungry than the traditional approaches that form the core contribution of this study.

**Keywords:** Movement disorders, pose-based analysis, NGLY1, SLC13A5, paediatrics

## 1. Introduction

Movement disorders encompass a range of neurological conditions that impair motor control and lead to symptoms such as tremors, ataxia, and involuntary movements. Clinical evaluation has traditionally relied on expert visual assessment in the clinic. Although such assessments are grounded in deep clinical expertise, they remain subjective, resource-intensive, and difficult to reproduce or scale. For ultra-rare paediatric conditions such as NGLY1 deficiency and SLC13A5 disorder, these limitations are particularly problematic because patients are geographically dispersed and often cannot attend frequent in-person evaluations.

Recent advances in computer vision and artificial intelligence (AI) have opened the possibility of extracting quantitative measurements from ordinary video recordings. Markerless pose-estimation algorithms such as OpenPose (Cao et al., 2019), VIBE (Kocabas et al., 2020), and PoseFormer (Zheng et al., 2021) can infer joint locations frame by frame, enabling the computation of spatio-temporal kinematic features without specialised motion-capture equipment. These methods have already been applied to conditions such as Parkinson's disease and cerebral palsy, providing objective measures of motor function that complement traditional scales.

In this study, we investigate whether such pose-based features can be used to quantitatively characterise movement disorders in paediatric patients with NGLY1 deficiency or SLC13A5 disorder. Because our cohort is very small and our clinical collaborators value interpretability, our main emphasis is on classical machine-learning models applied to carefully designed, physician-informed pose features. We aim to (i) define a set of intuitive, angle-based features that correlate with clinician severity ratings and distinguish between broad categories of movement disorder, and (ii) evaluate a panel of conventional classifiers on these features, highlighting the trade-offs between performance and interpretability. An important aspect of our design is that physician ratings are based on the original clinical videos, whereas the models are trained only on pose data extracted from those videos, since raw videos of paediatric patients are highly sensitive and cannot be readily shared across sites. As a secondary, exploratory analysis, we also adapt a transformer-based architecture that is pre-trained on a large public action-recognition dataset and fine-tuned on our clinical data. This experiment serves mainly to demonstrate what additional performance may be achievable with extensive pretraining.

## 2. Background and Related Work

### 2.1. Pose-based motion analysis in medicine

Markerless human pose estimation has transformed quantitative movement assessment. Deep-learning frameworks such as OpenPose extract 2D joint coordinates from video, allowing computation of spatio-temporal features such as joint angles and angular velocities without markers or wearable sensors (Cao et al., 2019). This approach has been successfully applied to a variety of clinical conditions. In Parkinson's disease, pose-derived trajectories have been used to quantify bradykinesia and tremor (Liu et al., 2019b); in cerebral palsy, similar methods capture abnormalities in spontaneous motor activity (Stenum et al., 2021), providing objective motor-function measures that complement traditional rating scales.

### 2.2. AI-driven assessment of movement disorders

Beyond simple feature extraction, AI models have been developed to detect and quantify movement disorders from pose data. In Parkinson's disease, pose-based features combined with Graph Neural Networks have classified tremor severity and assessed bradykinesia by highlighting subtle tremor movements that may be difficult to quantify by eye (Zhang et al., 2022; Quan et al., 2024). In gait analysis, pose-based metrics derived from joint trajectories, including step length, symmetry indices, and joint-angle entropy, have been used to identify gait abnormalities indicative of ataxia and other disorders (Tang et al.,

2022). For infant and developmental disorders, early detection systems for cerebral palsy have analysed spontaneous infant movements using pose-estimation models, showing that deviations from typical movement patterns can be detected from ordinary videos (Luo et al., 2022; Khan et al., 2018). Together, these studies show that pose-based analysis can yield clinically meaningful markers across a range of neurological conditions.

## 2.3. Transformers for motion modelling

A separate line of work has explored transformer architectures for modelling human motion. Pose Transformers (PoTr) (Martínez-González et al., 2021) introduced a non-autoregressive approach to motion prediction, thereby avoiding the error accumulation that can degrade performance in autoregressive models. Subsequent frameworks such as SPOTR (Nargund and Sra, 2023) and STPOTR (Mahdavian et al., 2023) disentangle spatial and temporal features, improving joint-trajectory prediction on large-scale motion datasets. Although these methods were developed primarily for general action recognition and motion synthesis, their ability to capture complex temporal dependencies suggests that, with sufficient data and careful pretraining, they could be applied to clinical video to assess diseases such as Parkinson's disease (Endo et al., 2022), for example by tracking disease progression or predicting changes in motor function. However, such models are typically less transparent than classical approaches and require larger datasets than are usually available in ultra-rare diseases.

## 2.4. Relevance to NGLY1 and SLC13A5 disorders

NGLY1 deficiency and SLC13A5 disorder are ultra-rare genetic syndromes characterised by poorly described motor delays and diverse movement-disorder phenotypes. Patients may exhibit both hyperkinetic movements (e.g. ataxia, chorea, myoclonus) and hypokinetic features (e.g. dystonia, bradykinesia), and these patterns can change with age. For such conditions, quantitative pose analysis offers three main advantages. First, it provides objective measurement: for example, tremor frequency and amplitude can be quantified directly from joint trajectories (Futrell et al., 2024), and similar principles can be extended to other movement patterns. Second, it enables longitudinal monitoring of therapy effects by tracking changes in movement features over time. Third, because these conditions are so rare and patients are widely distributed geographically, pose-based analysis can support remote assessments from home-recorded videos, reducing the need for frequent in-person visits.

Within this context, our primary goal is to understand whether a small set of intuitive, physically meaningful pose features can already capture clinically relevant differences between movement-disorder phenotypes in NGLY1 and SLC13A5. We therefore concentrate on traditional classification models, which can be inspected through feature importance and related tools, and treat deep transformer models as exploratory benchmarks.

## 3. Methods

Our pipeline begins by extracting 2D skeleton data (Figure 1), then derives a set of angular and joint-angle features from selected body parts, and finally applies a suite of classical

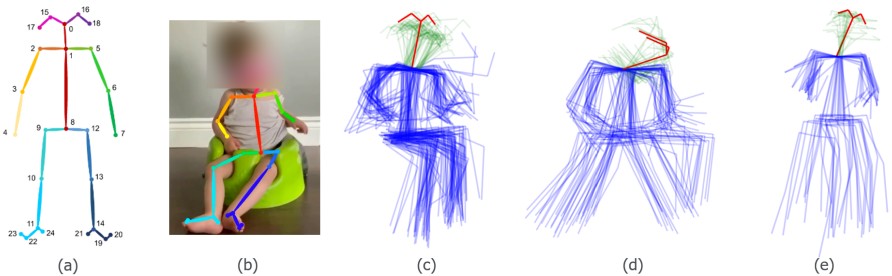

Figure 1: (a) OpenPose keypoint labels; (b) keypoints extracted from a representative frame; (c–e) keypoints extracted from video recordings from paediatric patients. The red lines highlight the mean neck angle.

| Angular Movement (keypoint labels) | Joint Angle (keypoint labels) |
|---|---|
| Head: (0,1) | Neck: (0,1,2), (0,1,5) |
| Upper limbs: (2,3), (5,6) | Shoulder: (1,2,3), (1,5,6) |
| Lower limbs: (9,10), (12, 13) | Elbow: (2,3,4), (5,6,7) |

Table 1: OpenPose keypoint labels of selected body parts.

classifiers to predict broad movement-disorder categories. In a separate, exploratory branch, we adapt a pre-trained pose transformer to the same task.

### 3.1. Pose features

The pose data are generated by the TRACER platform (Beneufit Inc.), which utilises the underlying OpenPose technology. The labels of the keypoints are given in Figure 1(a). Pose-based features derived from these keypoints are used to quantify the movement of a subject and to relate this movement to the severity of the movement disorder. In contrast to the clinicians, who scored the disorders directly from the original videos, all our models operate only on 2D skeleton sequences. This design choice reflects both privacy and data-governance constraints as pose sequences provide a de-identified representation that is more amenable to research use and potential future data sharing.

Body parts are selected based on the quality and consistency of the keypoints extracted by OpenPose. For the present analysis, we focus on head, upper limb, and lower limb segments. Table 1 summarises the keypoint pairs and triplets used to define angular-movement and joint-angle features.

#### 3.1.1. ANGULAR-MOVEMENT FEATURES

We segment the pose data extracted from each video into non-overlapping 10-frame intervals. For each segment, we compute the change in angle of a keypoint pair between the first and last frame of the segment. The mean angular displacement over $N$ segments is

$$\Delta\theta_{\mathrm{avg}} = \frac{1}{N} \sum_{i=0}^{N-1} \Delta\theta_i,$$

---

**Algorithm 1:** Computation of angular-movement features

---

**Input:** Sequence of keypoint coordinates $\{(x_t^{(a)}, y_t^{(a)}), (x_t^{(b)}, y_t^{(b)})\}_{t=0}^{T-1}$; segment length $L$ (e.g., $L = 10$ frames)

**Output:** Mean angular displacement $\Delta\theta_{\text{avg}}$ and variance $\sigma^2$

$N \leftarrow \lfloor (T - L)/L \rfloor$;

**for** $i \leftarrow 0$ **to** $N - 1$ **do**

$\quad | \quad t \leftarrow i \cdot L$;

$\quad | \quad \theta_t \leftarrow \arctan\left(\dfrac{y_t^{(a)} - y_t^{(b)}}{x_t^{(a)} - x_t^{(b)}}\right)$;

$\quad | \quad \theta_{t+L} \leftarrow \arctan\left(\dfrac{y_{t+L}^{(a)} - y_{t+L}^{(b)}}{x_{t+L}^{(a)} - x_{t+L}^{(b)}}\right)$;

$\quad | \quad \Delta\theta_i \leftarrow \theta_{t+L} - \theta_t$;

**end**

$\Delta\theta_{\text{avg}} \leftarrow \dfrac{1}{N} \sum_{i=0}^{N-1} \Delta\theta_i$;

$\sigma^2 \leftarrow \dfrac{1}{N} \sum_{i=0}^{N-1} (\Delta\theta_i - \Delta\theta_{\text{avg}})^2$;

---

and the variance is

$$\sigma^2 = \frac{1}{N} \sum_{i=0}^{N-1} (\Delta\theta_i - \Delta\theta_{\text{avg}})^2.$$

The angle $\theta_t$ at frame $t$ is defined from keypoints $a$ and $b$ as

$$\theta_t = \tan^{-1}\left(\frac{y_t^{(a)} - y_t^{(b)}}{x_t^{(a)} - x_t^{(b)}}\right).$$

Algorithm 1 summarises the computation in algorithmic form.

### 3.1.2. MEAN JOINT-ANGLE FEATURES

For joint-angle features, we compute the angle at a joint defined by three keypoints $p_t^{(1)}, p_t^{(2)}, p_t^{(3)}$ at each frame $t$. For instance, the shoulder angle uses neck, shoulder, and elbow keypoints. The angle is defined as

$$\theta_t = \cos^{-1}\left(\frac{(p_t^{(1)} - p_t^{(2)}) \cdot (p_t^{(3)} - p_t^{(2)})}{\|p_t^{(1)} - p_t^{(2)}\| \, \|p_t^{(3)} - p_t^{(2)}\|}\right),$$

and the mean joint angle over $T$ frames is given by

$$\theta_{\text{mean}} = \frac{1}{T} \sum_{t=0}^{T-1} \theta_t.$$

The corresponding algorithmic computation is summarised in Algorithm 2.

---

**Algorithm 2:** Computation of mean joint-angle features

---

**Input:** Sequence of keypoints $\{p_t^{(1)}, p_t^{(2)}, p_t^{(3)}\}_{t=0}^{T-1}$
**Output:** Mean joint angle $\theta_{\text{mean}}$
**for** $t \leftarrow 0$ **to** $T-1$ **do**
  $u_t \leftarrow p_t^{(1)} - p_t^{(2)};$
  $v_t \leftarrow p_t^{(3)} - p_t^{(2)};$
  $\theta_t \leftarrow \arccos\left(\dfrac{u_t \cdot v_t}{\|u_t\| \, \|v_t\|}\right);$
**end**
$\theta_{\text{mean}} \leftarrow \dfrac{1}{T} \sum_{t=0}^{T-1} \theta_t;$

---

## 3.2. Clinical labels and classification task

Pose data were extracted from videos of paediatric patients presenting with at least one movement disorder from ataxia, chorea, myoclonus, dystonia, hyperkinesia, or bradykinesia. For each video, two to three physicians independently scored the presence and severity of these disorders while being blinded to each other's assessments. Each movement disorder was rated on a five-point scale: Absent (0), Minimal (1), Mild (2), Moderate (3), and Severe (4).

All clinical ratings were performed on the original video recordings. However, for model development we did not use the raw pixel data: instead, each recording was first processed with OpenPose/TRACER to obtain 2D joint coordinates, and all features and classifiers were derived exclusively from these pose sequences.

For SLC13A5 patients, only ataxia, chorea, dystonia, and myoclonus scores were observed during physician assessment. For NGLY1 patients, we computed two composite scores: one combining ataxia, chorea, and myoclonus, and another combining dystonia, hypokinesia, and bradykinesia. Although these movement disorders have distinct clinical presentations, their symptoms can overlap, particularly when detailed clinical context is limited, making accurate differentiation challenging for both clinicians and machine-learning models. The limited size of our dataset further complicates robust model development.

To mitigate these challenges, we simplified the classification task into four categories defined by the predominant movement and coordination features. **Normometric** videos were those with absent or minimal movement disorder (all scores 0–1). **Hypometric** videos showed dystonia, hypokinesia, or bradykinesia with a score of at least 2. **Hypermetric** videos showed ataxia, chorea, or myoclonus with a score of at least 2. **Mixed-metric** videos exhibited both at least one hypometric feature (dystonia, hypokinesia, or bradykinesia) and at least one hypermetric feature (ataxia, chorea, or myoclonus). We then trained several machine-learning classifiers including Random Forest (RF), Multilayer Perceptron (MLP), Support Vector Machine (SVM), XGBoost (XGB), Logistic Regression (LR), and K-Nearest Neighbours (KNN), to predict these categories from the handcrafted pose features.

### 3.3. Transformer-based enhancement

In addition to the classical models, we trained a transformer-based model (Figure 2) in order to explore what performance could be achieved when leveraging large-scale pretraining. This analysis was not designed as an alternative clinical tool, but rather as an upper-bound estimate under more complex modelling.

The input to the transformer is a sequence of $t$ skeletons $\mathbf{X}_{1:t}$ extracted by OpenPose. During pre-training on a public action-recognition dataset, the network jointly learns to predict the next $M$ skeletons $\mathbf{X}_{t+1:T}$ and to classify the action class of each input sequence. In the subsequent fine-tuning stage on our clinical dataset, the model is optimised solely to categorise the movement disorder, while the motion-prediction branch is frozen to avoid overfitting.

Our architecture, adapted from Martínez-González et al. (2021); Mahdavian et al. (2023), consists of several interconnected modules. A Graph Neural Network (GNN) encoder $\phi$ maps each input skeleton $\mathbf{x}_t$ to a fixed-dimensional embedding. Positional embeddings are then added to these skeleton embeddings before they pass through $L$ multi-head self-attention layers in the transformer encoder, yielding a latent representation $\mathbf{z}_{1:t}$. A linear classification head processes $\mathbf{z}_{1:t}$ to produce either action class logits (during pre-training) or movement-disorder logits (during fine-tuning).

Simultaneously, the transformer decoder takes as input the encoder outputs $\mathbf{z}_{1:t}$ along with a query sequence $\mathbf{q}_{1:M}$ initialised to the last observed skeleton $\mathbf{x}_t$. After $L$ layers of self-attention, the decoder outputs a sequence of embeddings which the decoding network $\psi$ transforms into reconstructed future skeletons $\hat{\mathbf{X}}_{t+1:T}$.

The overall loss during pre-training is the sum of a classification term and a motion reconstruction term:

$$L_{\text{total}} = L_{\text{cls}} + L_{\text{motion}}.$$

Here, $L_{\text{cls}}$ denotes the cross-entropy loss for action classification. The motion loss $L_{\text{motion}}$ is computed by averaging layerwise $\ell_1$ reconstruction errors across all decoder layers. If $\hat{\mathbf{y}}_m^l$ is the predicted $N$-dimensional pose vector at time step $m$ in decoder layer $l$, and $\mathbf{y}_m^*$ is the ground truth, then each layer's loss is given by

$$L_l = \frac{1}{M \cdot N} \sum_{m=t+1}^{T} \left\| \hat{\mathbf{y}}_m^l - \mathbf{y}_m^* \right\|_1,$$

and $L_{\text{motion}} = \frac{1}{L} \sum_{l=1}^{L} L_l$. In the fine-tuning stage, only the classification head is updated on the clinical labels, and the transformer results are reported primarily as a point of comparison with the more interpretable classical models.

## 4. Experimental Setup

### 4.1. Dataset

Our dataset comprises 95 video recordings of 26 paediatric patients with NGLY1 deficiency or SLC13A5 disorder, captured in standing or sitting positions. Each patient was recorded at different ages during their disease progression, and 13 video recordings were excluded from the analysis because the subjects were too young (less than 2 years old) to stand or sit

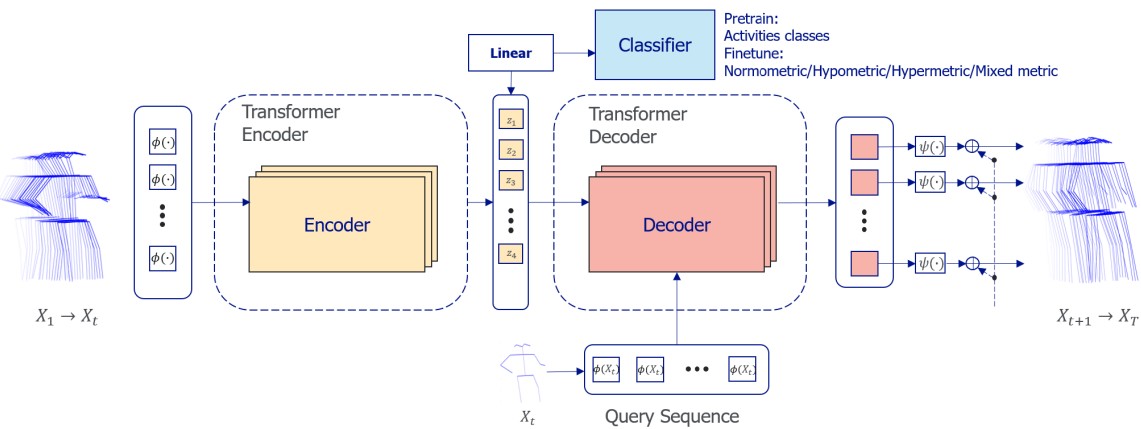

Figure 2: Transformer-based framework for predicting motion and movement disorder.

independently at the time of recording. The raw videos were used only for clinical review and for extracting 2D skeletons. The mean age of the subjects in the analysed recordings is $9.86 \pm 4.50$ years. The mean number of frames per recording is $1259.08 \pm 131.52$. After filtering, the dataset contains 7 normometric samples, 11 hypometric samples, 40 hypermetric samples, and 25 mixed-metric samples. Pose data are normalised using the neck (keypoint label 1) as the root with all joint coordinates translated so that the root is fixed at $(0, 0)$ while the relative distances between joints remain unchanged. To pre-train the transformer, we also used 5,688 videos (standing, sitting, staggering gait, etc.) from the NTU RGB+D 120 dataset (Liu et al., 2019a). For fine-tuning, each pose data sample was downsampled to 100 frames to match the non-overlapping 10-frame intervals used for deriving the handcrafted features.

## 4.2. Model training and evaluation

The prepared data were partitioned into an 80% training set and a 20% test set using stratified sampling to preserve the class distribution. For the classical machine-learning models (RF, MLP, SVM, XGB, LR, and KNN), a comprehensive feature selection process was conducted on training set. First, an ensemble selection method ranked the features using the Chi-squared test, Mutual Information, ANOVA F-test, and Recursive Feature Elimination. Subsequently, the final feature set was determined by a voting system, where features ranked in the top 12 by at least three of the four methods were selected for modelling. All features were standardised based on the training data.

To address the imbalanced nature of the dataset, class weighting was applied. The training process was conducted in two phases. Initially, all models were evaluated using a 5-fold stratified cross-validation on the training set to establish baseline performance based on the weighted F1-score. Following this, the top three performing models underwent extensive hyperparameter tuning via grid search, which also utilised a 5-fold stratified cross-validation strategy. Finally, all models, including the optimised versions, were assessed on the held-out test set.

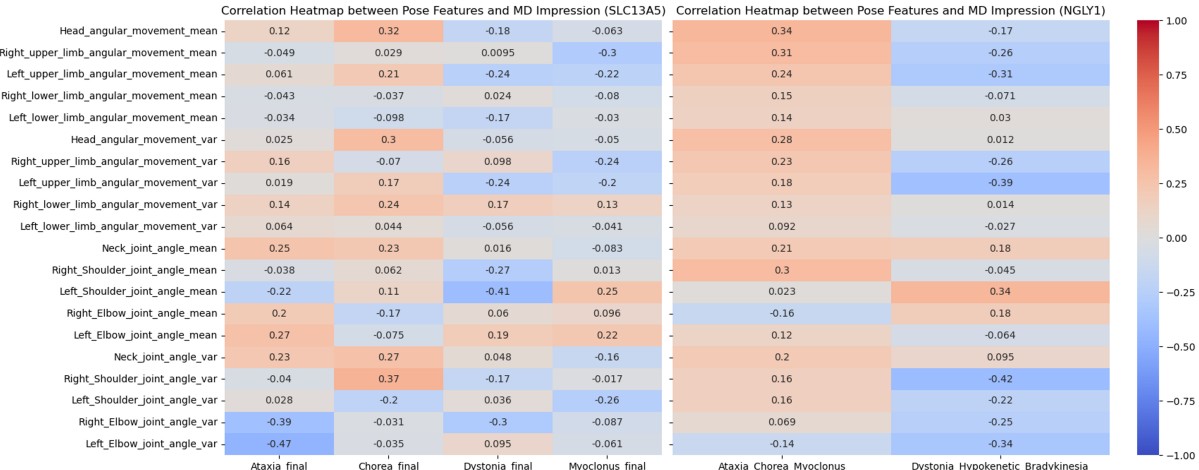

Figure 3: Correlation between clinical severity scores and pose-based features.

The transformer-based model was pre-trained on the NTU RGB+D 120 dataset for 100 epochs with an initial learning rate of $1 \times 10^{-4}$, then fine-tuned on our training set. Due to limited patient data, we froze the motion-prediction branch during fine-tuning and trained only the classification head for 50 epochs at the same learning rate.

## 5. Results and Discussion

We first examined the correlation between clinician assessment scores and handcrafted features (Figure 3). Although the correlation coefficients are modest, pose-based features consistently show positive associations with clinical severity ratings, supporting their effectiveness as quantitative proxies for clinician-observed movement abnormalities.

For the classification task, we assessed model performance using accuracy, weighted precision, recall, and F1-score. The RF classifier achieved the highest accuracy among the classical models at 65%, while other models such as XGB and LR achieved slightly lower but comparable performance(Table 2). The top five features used by the RF model include the mean and variance of left upper-limb angular movement, the mean and variance of neck angle, and the mean of head angular movement. The upper limbs and neck/head are also considered particularly important by physicians when assessing patients. Given the limited patient data, these results are encouraging, especially in light of the fact that RF and related models can provide feature importance estimates that help clinicians understand which aspects of movement are most discriminative across categories. The fine-tuned transformer-based model achieved the same accuracy as RF but obtained higher recall and F1-score due to improved performance in predicting the mixed-metric class. The transformer-based model achieved an accuracy of 83% for the mixed-metric class, compared with 50% for RF, highlighting its ability to identify more complex manifestations of movement disorders. This improvement illustrates the potential of large-scale pretraining, but it comes at the cost of reduced interpretability and greater complexity, and depends critically on access to extensive non-clinical training data.

| Model | Precision | Recall | F1-score | Accuracy |
|---|---|---|---|---|
| RF | **0.72** | 0.57 | 0.62 | **0.65** |
| MLP | 0.60 | 0.50 | 0.55 | 0.47 |
| SVM | 0.65 | 0.55 | 0.60 | 0.53 |
| XGB | 0.70 | 0.55 | 0.62 | 0.59 |
| LR | 0.51 | 0.59 | 0.50 | 0.59 |
| KNN | 0.60 | 0.50 | 0.55 | 0.47 |
| Transformer | 0.71 | **0.60** | **0.64** | **0.65** |

Table 2: Classification results. The transformer is pre-trained on NTU RGB+D and fine-tuned on patient data; classical models are trained only on clinical features.

## 6. Conclusion

We have presented a quantitative, AI-driven framework for assessing movement disorders in paediatric patients with NGLY1 deficiency and SLC13A5 disorder, focusing on interpretable pose-based features and traditional machine-learning models suitable for small datasets. Our framework deliberately decouples clinical assessment (performed on raw videos) from model training (performed on de-identified pose data), which better aligns with the privacy constraints inherent to paediatric ultra-rare disease cohorts and increases the feasibility of future cross-centre data sharing. Our results show that these simple, physician-informed features can differentiate between broad movement-disorder categories and correlate with clinician severity ratings. An exploratory transformer experiment demonstrates that higher predictive performance is possible when leveraging large-scale pretraining, but such models are less transparent and more difficult to deploy in routine clinical practice.

Future work will focus on refining our feature set, incorporating larger and more diverse clinical cohorts, and explicitly integrating clinician feedback on interpretability and usability. In the longer term, we envisage combining classical models for day-to-day use with more complex deep learning models as research tools to explore subtle patterns in larger, multi-centre datasets. Our ultimate goal is to deliver a reliable, objective assessment tool that supports clinicians in diagnosing and monitoring these rare paediatric movement disorders while respecting the practical constraints of ultra-rare disease research.

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
