# OpenReview forum: "Quantitative Pose-Based Analysis of Movement Disorders in Pediatric NGLY1 and SLC13A5 Patients"
_MIDL.io/2026/Conference — MIDL 2026 Poster_

### Official Review · Reviewer_Y61x · 2026-01-08

**Confidence:** 4
**Preliminary Rating:** 3
**Final Rating:** 3

**Summary:**

This paper proposes a markerless pose-based pipeline to classify movement disorders in children with rare NGLY1 and SLC13A5 mutations. Using OpenPose, the authors extract 2D skeletons from 95 videos and derive interpretable geometric features (e.g., joint angles) for classification via Random Forest (RF) and other classical models. The paper compares these interpretable models against a pre-trained Transformer baseline. Results show the RF model achieves comparable accuracy to the Transformer, but with better interpretability for clinical use.

**Strengths:**

- Clinical Utility: The work addresses a crucial need for objective, remote monitoring tools for ultra-rare pediatric diseases. The correlation analysis (Figure 3) validates that the "handcrafted" features align meaningfully with clinician severity ratings.

- Interpretability: The prioritization of physician-informed features (e.g., neck/limb angles) over black-box deep learning is a strong design choice for this domain, fostering trust with clinical collaborators.

- Benchmarking: The inclusion of a Transformer baseline (pre-trained on NTU RGB+D) provides a useful performance upper bound, highlighting the trade-offs between complexity and explainability on small datasets.

**Weaknesses:**

- Dated 2D Pose Estimation: The pipeline relies on OpenPose (2019), which is no longer state-of-the-art. The authors do not justify this choice over more robust, modern frameworks like MMPose, ViTPose, DeepLabCut or HRNet, nor do they discuss alternatives, such as 3D pose estimation models or mesh models.

- Ambiguous Validation Strategy: The paper states data were partitioned using "stratified sampling" but does not explicitly confirm if this was done at the patient level. Given the small sample size (N=95 videos from 26 patients), if multiple videos from the same patient appear in both train and test sets, the results are likely inflated due to identity leakage.

- Lack of Granular Results: Table 2 aggregates performance into a single accuracy metric. Since the "Mixed-metric" class is notably difficult (RF accuracy 50% vs Transformer 83%), the lack of a per-class breakdown (Precision/Recall/F1 for each phenotype) obscures failure cases.

- Missing Ablation: The feature selection process used a "voting system", but there is no ablation study to show which body parts (e.g., upper vs. lower limb) contribute most to the performance.

**Detailed Comments:**

- Presentation: Please expand Table 2 to include metrics per class. The current average F1-score hides the model's inability to handle complex phenotypes.

- Context: The "Background" section should acknowledge broader advances in human mesh recovery (HMR) or animal pose estimation (e.g., DeepLabCut, SLEAP) that are highly relevant to quantitative behavioral analysis.

- Phrasings: The abstract describes OpenPose as a "modern computer vision tool".

- Citations:
- - for Clinical Scales: In the Introduction, you mention "physician-informed pose features". Please cite the specific clinical rating scales that inspired these features.
- - for application of kinematic features: In the Introduction, you mention that "...been applied to conditions such as Parkinson's disease and cerebral palsy". Please cite the corresponding studies.

**Justification Of Final Rating:**

I thank the authors for addressing the rebuttal questions. While the paper addresses an interesting clinical problem with a focus on interpretability, I remain concerned about the limited technical execution due to the reliance on outdated pose estimation frameworks without sufficient comparison to alternatives. Additionally, the lack of patient-independent validation, limited technical novelty, and absence of code sharing raise concerns about reproducibility and generalization. Hence, I keep my current score.

**Justification Of The Preliminary Rating:**

The study addresses a worthy clinical problem with a sound emphasis on interpretability. However, the technical novelty is limited by the use of dated pose estimation tools, without comparison to modern alternatives, and handcrafted features. The ambiguity regarding subject-independent validation is critical; the authors are asked to clarify data splits and provide more detailed performance metrics.

**Questions To Address In The Rebuttal:**

**Data Split Independence:** Was the train/test split performed strictly at the subject level? If not, please provide an evaluation on unseen patients to prove the model isn't overfitting to subject identity.

**Per-Class Metrics:** Please provide the F1-scores and confusion matrices for each of the four specific classes (Normometric, Hypometric, Hypermetric, Mixed).

**Modern Baselines:** Why was OpenPose chosen over newer, more robust estimators? Have you validated that OpenPose accurately tracks pediatric limbs during occlusions?

**Feature Importance:** Which specific features were selected by the voting system? A list of the top 5 discriminatory features is needed to support the interpretability claims.

---

> ### Author Response · Authors · 2026-01-23
> **Response to Reviewer Y61x**
>
> We thank the reviewer for their careful reading and detailed comments. We have revised the manuscript accordingly and respond point-by-point below.
>
> **Dated 2D pose estimation (OpenPose)**
>
> We agree that OpenPose is no longer state-of-the-art. In this study, pose extraction was performed using the TRACER platform provided by our collaborator Beneufit, which was the agreed and validated processing pipeline for these clinical recordings when the project was initiated. TRACER provides a consistent and standardized preprocessing step across all videos in this cohort and includes proprietary robustness and quality-control components. OpenPose itself has also been validated against gold-standard marker-based systems in pediatric settings [1][2].
>
> Importantly, our downstream pipeline (feature computation, feature selection, and interpretable modeling) is largely pose-estimator agnostic: the same analysis can ingest keypoints from other estimators (e.g., HRNet, ViTPose) without changing the modeling logic. We now add a dedicated discussion acknowledging newer 2D estimators and emerging 3D pose/mesh recovery methods and position benchmarking alternative estimators as future work.
>
> **Validation strategy and possible identity leakage**
>
> Thank you for raising this important point. In our current experiments, stratified sampling was performed at the video level to preserve the highly imbalanced distribution of the four phenotype classes. We did not enforce a strict patient-level split. We agree this can lead to optimistic estimates if recordings from the same child appear in both train and test sets.
>
> Our dataset contains repeated recordings separated by 4 to 24 months, and phenotypes can change substantially with development and disease progression. Therefore, the video-level split reflects a within-cohort monitoring scenario rather than a strict “generalization to unseen patients” setting. We revise the manuscript to (i) state the splitting strategy unambiguously, and (ii) include this as a key limitation.
>
> **Granular results**
>
> Thank you. We now include per-class precision/recall/F1 and confusion matrices, which make failure modes (especially Mixed-metric) explicit.
>
> **Feature importance / ablation**
>
> Thank you. We add (i) the selected features (highlighted in the correlation plot) and (ii) RF feature importance analysis to support interpretability claims, and we discuss which body regions contribute most strongly.
>
> **Additional context and citations**
>
> Thank you. We expanded the Background section to acknowledge broader advances in pose and mesh recovery and clarified phrasing in the abstract (e.g., avoiding calling OpenPose “modern”). We also add clinical-scale citations motivating the physician-informed feature design and add citations for medical applications of kinematic/pose features.
>
> [1] Anderson, Jeffrey T., et al. "Validation of markerless video-based gait analysis using pose estimation in toddlers with and without neurodevelopmental disorders." Frontiers in Digital Health 7 (2025): 1542012.
>
> [2] Washabaugh, Edward P., et al. "Comparing the accuracy of open-source pose estimation methods for measuring gait kinematics." Gait & posture 97 (2022): 188-195.

---

### Official Review · Reviewer_ZMaM · 2026-01-09

**Confidence:** 4
**Preliminary Rating:** 3
**Final Rating:** 5

**Summary:**

This work uses markerless computer vision (OpenPose) to extract interpretable pose features from video recordings and classify abnormal movements in children with NGLY1 or SLC13A5 mutations. Focusing on simple, clinically meaningful features and conventional classifiers, the study shows that these handcrafted measures can distinguish movement-disorder phenotypes even with very small datasets.

Experiments show that the pre-trained, fine-tuned transformer suggests higher potential performance. The authors state that such deep models are less interpretable and require more data than the traditional, feature-based approaches central to the study.

**Strengths:**

The authors used two stage strategy to achieve pediatric disorders: pose estimation and spatial-temporal classification which is better than end-to-end deep learning classifier. The paper is well written and easy to follow. The authors use real patient data, enhancing clinical relevance and authenticity.

**Weaknesses:**

Since transformer is the best model among all baselines, there are still some missing parts in this paper:
1. What is the size of the transformer, even though it's cited, I'd like the authors to explicitly list it in the paper. What computing resources do the authors have.
2. Is there any augmentation applied during the training?
3. Based on the final results, it's not very convincing that it worths using transformer than RF. I understand data might be an issue.
4. Ablation studies are limited. For example, why choose X_T as the query input.

**Detailed Comments:**

1. Please clearly explain the size of the transformer and computing resources.
2. How accurate is the first pose estimation module? What is the impact of it.
3.  Is there any augmentation applied during the training?
4. Why choose X_T as the query input. Could it be better to use other frames / latent features as query input?
5. There similar applications and methodologies in fetal MRI, https://arxiv.org/abs/1907.04500 and https://arxiv.org/abs/2007.08146. The authors can add them in the introduction part.
6. In your title, it should be Pediatric not Paediatric.

**Justification Of Final Rating:**

Thank the authors for the rebuttal efforts.
My concerns are addressed and I understand that the focus of this work is to explore a systematic pipeline for disorders classifications. Transformer is compared as a baseline method.
I raised my rating to accept.

**Justification Of The Preliminary Rating:**

The authors demonstrated their method on a real clinical dataset and explored the use case of transformer. There are many non-dl based baselines and transformer appears to be the best among them. However, there are missing details and lack of ablations which limits the quality of the paper. I understand it might be due to the page limitation and I'd like to hear more from the authors during rebuttal.

**Questions To Address In The Rebuttal:**

I list the questions I have in the detailed comments. I'd like to hear more details from the authors.

---

> ### Author Response · Authors · 2026-01-23
> **Response to Reviewer ZMaM**
>
> We thank the reviewer for the constructive feedback and helpful suggestions. We address each point below.
>
> **Transformer size and computing resources**
>
> Thank you for pointing this out. The transformer is adapted from the PoTr-style [1] framework and uses 4 encoder layers and 4 decoder layers, with FFN dimension 2048. Training was performed on an NVIDIA L40S GPU; with batch size 32, peak VRAM usage was approximately 4 GB. We have added these details to the revised manuscript.
>
> **Data augmentation**
>
> No data augmentation was used for training the transformer.
>
> **Whether the transformer is “worth it” vs RF**
>
> Thank you for raising this important point. We agree that, given the small dataset and the need for clinical interpretability, Random Forest (RF) and other classical models are the most practical focus of this work. Our intent in including the transformer is not to argue for near-term deployment, but rather to provide a benchmark indicating what performance might be achievable with extensive pretraining. We have also added confusion matrices and per-class metrics to show that the transformer performs better on the Mixed-metric class, which could be useful in future studies as larger datasets become available.
>
> **Limited ablations (e.g., why use $X_t$ as the query)**
>
> Thank you for pointing this out. We use the last observed skeleton $X_t$ as the query because it is the most recent known state available at inference time and provides a stable anchor at the boundary between observed and predicted motion, which is consistent with prior work. We now clarify this design choice in the Methods section.
>
> **Pose estimation accuracy / impact**
>
> Thank you for raising this. In our study, pose extraction is performed using the TRACER platform, which is based on OpenPose with additional preprocessing steps to produce standardized keypoint sequences used for all downstream analyses. TRACER was the agreed-upon and validated processing pipeline for our clinical recordings when the project was initiated. It is designed for objective movement analysis from smartphone videos in real-world settings, has been used across multiple clinical projects by our collaborators, and provides a consistent and standardized preprocessing step for all videos in this cohort. OpenPose itself has also been validated against gold-standard marker-based systems [1][2]. Crucially, we treat pose estimation as a fixed upstream module, and our downstream pipeline (feature computation, feature selection, and interpretable modeling) is largely pose-estimator agnostic. That said, pose-estimation quality is critical to classification performance.
>
> **Related work in fetal MRI**
>
> We agree that these works are relevant. We added the cited fetal MRI works to the Related Work section.
>
> **Title spelling**
>
> Thank you. We have switched to US spelling throughout (e.g., “Pediatric”, "Standardize").
>
> [1] Anderson, Jeffrey T., et al. "Validation of markerless video-based gait analysis using pose estimation in toddlers with and without neurodevelopmental disorders." Frontiers in Digital Health 7 (2025): 1542012.
>
> [2] Washabaugh, Edward P., et al. "Comparing the accuracy of open-source pose estimation methods for measuring gait kinematics." Gait & posture 97 (2022): 188-195.

---

### Official Review · Reviewer_9qrA · 2026-01-09

**Confidence:** 4
**Preliminary Rating:** 3
**Final Rating:** 4

**Summary:**

This contribution is concerned with two rare diseases, two genetic mutations that lead to developmental movement problems. Some authors are from a company that develops a commercial application that allows to analyse movement patterns based on marker-free pose estimation from simple mobile phone video footage (based on OSS DL models plus custom analysis), which is also the topic of this paper. In this work, they authors first extract pose information, then compute simple hand-crafted movement features from a sequence of frames and

**Strengths:**

This paper clearly targets a clinical application and is well-motivated. I would consider the fact that it targets rare diseases neutral – on the one hand, the impact is limited when considering global population, on the other hand, such impact is *huge* for those concerned, who are often left alone due to limited commercial interest. It is definitely an interesting task.

**Weaknesses:**

* It is not really a weakness of the paper per se, but it needs to be decided how well this contribution fits the scope of the MIDL conference, since the focus is hand-crafted features, not on deep learning. Since the results are compared against a DL-based classifier and the original pose information is extracted with a DL method, I would personally conclude that it is interesting enough for MIDL, although the only actual trainings on the pediatric dataset are either classifical methods (random forest, SVM, XGBoost etc.) or just a single linear layer on top of a pre-trained transformer architecture.
* The dataset is relatively small (I wonder why no control group was included?) and I wonder if the differences in the results are statistically significant.
* The best interpretable classifier, which the authors apparently preferred over the Transformer-based model achieved only 65% accuracy. It is not clear to me how useful this would be already in the context of the targeted disorders.
* The paper allocates quite some space to trivial algorithms, but is quite terse later, when describing the feature analysis. This could be improved during a revision, I think.
* While I agree that the classical classifiers are interpretable, the features appear less intuitive to me: Doctors would probably not take the stddev of the angulation change in a temporally downsampled sequence (every 10th frame) into account. One could also argue that the linear layer trained as an alternative is *more* interpretable than an MLP (the stress being on the M), so the underlying features need to be considered as well. It also appears to me that the features are not invariant towards view direction, but it is not discussed whether all videos were taken from the front / back, for instance.

**Detailed Comments:**

* I interpret $\Delta \theta_i$ to be the "change in angle of a keypoint pair between the first and last frame of the segment"; if that's correct, I suggest to name it in the sentence as such.
* Algorithms 1 and 2 on the other hand are not really necessary; they takes a lot of space while not providing more information than the precisely worded explanation. Speaking of "does not hurt", do you really use arctan and subtraction? Or arctan2 and some angle-aware subtraction? I guess you also compute the angles only once and not twice while iterating. Getting rid of the algorithms and ensuring that the mathematical definitions are precise prevent such questions.
* I find it strange that three references for markerless pose-estimation are given in the introduction, but none for its medical applications, then in 2.1 only one reference is given for the former, but two for the latter. I suggest to make that consistent.
* For Table 1, I suggest to look into the "booktabs" package (and maybe left alignment?) for nicer formatting.
* Speaking of formatting, I suggest to use unbreakable spaces (~) between "Severe" and "(4)" etc.
* Interpretation of the results could be easier if you also presented 4x4 confusion matrices (e.g., for the Transformer architecture and the RF).
* I guess with "features were standardised" you mean whitening? I suggest to slightly elaborate.
* Please explain what exactly you mean with "downsampled to 100 frames". Given that the number of frames per recording varies, but you chose a fixed frame distance (10) for the classical features, there is room for interpretation here. (And if you did not use equidistant sampling, I wonder why you make this huge change from the feature extraction approach.)

**Justification Of Final Rating:**

The manuscript has been improved considerably.  Still, there are a number of things that look unfinished in the manuscript, and in light of the strong competition, I think this is actually somewhere between "borderline" and "weak accept".
The small dataset is due to the rare disease and pediatric domain, so that should not speak against this paper, although it makes the quantitative evaluation inherently weaker.
Some statistical hypothesis testing was introduced now, but apparently after the fact and incompletely (no discussion, no p-value cut-off defined, and the number of tests performed – 36 in Figure 5 alone – would then also motivate a multiple testing correction).
(If accepted, before publication, Tables 2 and 3 should be formatted similar to 1 as well.)

**Justification Of The Preliminary Rating:**

I would have suggested to accept this paper because I find the work interesting, but the manuscript currently still has a number of weaknesses as it is now. If the revision manages to address the points I listed above, my current impression is that I would rate it "weak accept".

**Questions To Address In The Rebuttal:**

Can you elaborate in the paper how separating the four target classes helps clinically? How do these classes relate to the two diseases? It appears to me that the diseases cannot be easily separated from each other? How about a comparison with a healthy control? What would be a good operating point?
Also, please respond to the feature / interpretability discussion above (last item in "weaknesses").

---

> ### Author Response · Authors · 2026-01-23
> **We thank Reviewer 9qrA for considering this paper interesting.**
>
> **Scope / fit for MIDL (handcrafted features vs deep learning)**
>
> Thank you for raising this. While our downstream models on the pediatric cohort are primarily classical (to match the small-
> 𝑁, clinician-interpretability setting), the study still aligns with MIDL because it investigates a deep-learning–enabled clinical pipeline end to end. First, a validated DL-based pose estimator converts real-world smartphone videos into standardized kinematic signals, enabling quantitative phenotyping in ultra-rare pediatric cohorts where large curated datasets are impractical. Second, we include a pretrained pose-transformer benchmark to contextualize what additional performance might be achievable with large-scale pretraining, while explicitly highlighting the interpretability–complexity trade-off that motivates our focus on classical models. We believe this combination of clinically grounded problem, ML-enabled measurement from real-world video, and careful discussion of deployment constraints fits MIDL’s scope.
>
> **Small dataset, lack of control group, and statistical significance**
>
> Thank you for this comment. We agree the cohort is small, reflecting (i) the ultra-rare nature of both conditions and (ii) practical and ethical constraints in acquiring and processing pediatric clinical videos at scale. We did not have access to an age-matched healthy pediatric cohort collected under the same protocol. To provide context, we performed an exploratory comparison against a subset of the NTU dataset as a reference. However, NTU participants are generally older and recordings differ in acquisition conditions, so we treat this as a reference/sanity check rather than a matched control analysis.
>
> We have added significance testing for feature differences between the patient cohort and the NTU reference subset and report the results in the revised manuscript.
>
> **Utility of 65% accuracy for interpretable models**
>
> We agree that overall accuracy (65%) must be interpreted in the context of (i) a small, heterogeneous ultra-rare cohort and (ii) the intrinsic difficulty of the Mixed-metric category, which represents co-occurring hyperkinetic and hypokinetic features and is expected to overlap with other classes. Importantly, our error analysis shows the RF perform substantially better at identifying Hypermetric phenotypes, and most errors arise from Mixed-metric samples.
>
> Clinically, our intended use is not autonomous decision-making, but standardized phenotype summarization from real-world videos to support longitudinal monitoring and reduce inter-rater variability. In this setting, reliably distinguishing predominantly hypo- vs hypermetric movement can already be useful for tracking progression and treatment response, while Mixed-metric predictions can be treated as “overlapping/uncertain” cases. We now include per-class metrics and confusion matrices to make this clearer. We also demonstrate the power of transformer for identifying Mixed-metric, which can potentially address complex manifestations in the future.
>
> **Feature interpretability, temporal downsampling, model choice, and view invariance**
>
> We agree that interpretability needs to be considered at both the model level and the feature level. Our goal is not that clinicians would explicitly compute a standard deviation or a variance, but that this statistic serves as a quantitative proxy for clinically observed constructs such as rhythm/regularity and intermittency/hesitation. Concretely, the standard deviation or variance captures variability in movement speed and excursion across the recording, which clinicians often recognize in phenomena such as bradykinetic hesitations or irregular motion.
>
> We use a fixed 10-frame window as temporal smoothing to reduce pose noise and focus on gross movement irregularity.
>
> Finally, you are correct that 2D angle-based features are not inherently view-invariant. In our data-collection protocol, parents were instructed to record children front-facing as much as possible to standardize viewpoint.
>
> **Detailed comments**
>
> - Algorithms 1–2 and angle computation details
>
> This is a good point. We removed the algorithms and strengthened the mathematical definition. We also clarify the implementation: we compute angle changes using a dot-product / norm formulation.
>
> - Pose-estimation references vs medical applications
>
> We reorganized the Introduction and Related Work to improve coherence.
>
> - Table formatting
>
> We added nonbreaking spaces.
>
> - Confusion matrices
>
> We include confusion matrices for RF and the transformer.
>
> - “Standardized features” clarification
>
> We used z-scoring.
>
> - "Downsampling" ambiguity
>
> In our implementation, preprocessing is consistent across the classical models and the transformer. Specifically, we trim each recording under neurologists supervision to remove less informative lead-in/lead-out segments and retain a fixed-length clip. We then apply the same uniform fixed-stride sampling (every 10th frame) for both approaches.

---

### Author Response · Authors · 2026-01-23
**Author Response - General**

We sincerely thank all reviewers for their thoughtful and constructive feedback. We are encouraged by the positive assessment of the clinical relevance and the emphasis on interpretability for ultra-rare pediatric movement disorders. In response to the reviews, we substantially revised the manuscript to improve clarity, cohesion, and completeness, and to better align the paper’s central message with the small-𝑁, clinician-facing setting.

Across the revision, we made the following key improvements:

- **Stronger narrative and positioning** We reorganized the Introduction and Related Work to sharpen motivation, and clearly state the main contribution: interpretable, physician-informed pose features + conventional classifiers for small datasets, with the transformer included only as an exploratory benchmark to contextualize potential performance under large-scale pretraining.

- **Clearer and more precise methods** We removed low-value algorithm blocks and replaced them with concise, unambiguous definitions and implementation details. In particular, we clarify how angular/joint-angle quantities are computed in practice (dot-product / norm-based formulations), reducing ambiguity around $\arctan$ subtraction and wrap-around behavior.

- **Transparent preprocessing and sampling** We clarified that the same trimming and uniform fixed-stride sampling are used consistently across classical models and the transformer, and we explain the rationale (standardized clips, reduced noise, stable training/evaluation).

- **More complete reporting of results** We added per-class precision/recall/F1, confusion matrices, and expanded error analysis to make failure modes explicit (especially for the Mixed-metric class).

- **Statistical evidence for feature differences** We added formal significance testing for feature differences (patient cohort vs reference subset), while clarifying that the reference dataset is not an age-matched pediatric control and is therefore used as contextual comparison rather than a matched control group.

- **Interpretability strengthened** We report the most discriminative features (including feature-selection outcomes and model-based importance summaries where applicable) and connect them to clinically meaningful movement characteristics, while acknowledging that interpretability depends on both model and feature design.

- **Clarifications on pose estimation choice and limitations** We clarified that pose extraction is performed via the TRACER pipeline (based on OpenPose with proprietary robustness steps), and we explicitly acknowledge limitations of relying on a fixed upstream estimator. We also note that the downstream framework is largely estimator-agnostic and can ingest keypoints from newer systems in future work.

- **Evaluation protocol clarified and limitations stated** We explicitly state the splitting strategy used in the current experiments and limit claims accordingly, acknowledging the potential for non-independence in video-level splits when repeated recordings exist. We add this as a clear limitation and motivate subject-independent evaluation as future work.

We believe these revisions directly address the reviewers’ main concerns, especially around manuscript cohesion, methodological precision, completeness of experimental reporting, and clearer interpretation and limitations, while preserving the practical clinical focus of the work. We are grateful for the reviewers’ guidance, which materially improved the paper.

---

### Author Rebuttal · Authors · 2026-01-24

**Rebuttal:**

We revised the manuscript in response to the reviewers’ comments. All changes are highlighted in the revised submission and summarized in the “Author Response – General” section. We appreciate the reviewers’ time and feedback and look forward to further discussion.

**Supporting Material:**

/attachment/f6800e94dc4935226b7050efdc71df609d2cdf57.pdf

---

### Meta-Review · Area_Chair_amtR · 2026-02-05

**Recommendation:** Reject
**Confidence:** 5

**Metareview:**

While the paper addresses a clinically relevant problem in an ultra-rare pediatric population and the authors have made a visible effort to revise the manuscript, the current experimental evidence does not appear sufficient to support acceptance at MIDL.
The main concerns are methodological. The evaluation protocol relies on video-level stratified splits despite the presence of multiple recordings per patient. As acknowledged by the authors, this setup introduces a substantial risk of identity leakage, which makes the reported accuracy, F1-scores, and model comparisons difficult to interpret as indicators of generalization. Without a subject-independent evaluation strategy, the quantitative results cannot reliably support claims about classification performance, even in an exploratory context.
In addition, the comparison between classical models and the transformer-based approach remains insufficiently supported. The transformer is trained with a frozen encoder and limited fine-tuning, and no ablation studies or uncertainty estimates are provided. The reported performance differences relative to RF baseline are small and not accompanied by variance or confidence intervals, making it unclear whether these differences are meaningful.
More generally, the statistical reporting remains limited given the small and imbalanced dataset. Although the rebuttal indicates that per-class metrics and confusion matrices were added, these elements are not present in the submitted manuscript. Table 2 reports only aggregate metrics, which obscures class-specific behavior and prevents a clear assessment of model failure modes.
In summary, while the clinical motivation and emphasis on interpretability are clear strengths of the work, the current experimental design and reporting fall short of the level of rigor expected for quantitative evaluation at MIDL.

---

### Decision · Program_Chairs · 2026-02-13

Accept (Poster)